# Age at Arrival and Depression among Mexican Immigrant Women in Alabama: The Moderating Role of Culture

**DOI:** 10.3390/ijerph19095342

**Published:** 2022-04-27

**Authors:** Courtney Andrews, Kathryn S. Oths, William W. Dressler

**Affiliations:** 1Institute for Human Rights, University of Alabama at Birmingham, Birmingham, AL 35294, USA; 2Department of Anthropology, The University of Alabama, Tuscaloosa, AL 35487, USA; koths@ua.edu (K.S.O.); wdressle@ua.edu (W.W.D.)

**Keywords:** cultural consonance, acculturation, immigration, depression, mental health, Mexican immigrant women

## Abstract

Mexican-born women in the U.S. are at high risk of depression. While acculturation is the primary analytical framework used to study immigrant mental health, this research suffers from (1) a lack of specificity regarding how cultural models of living and being take shape among migrants converging in new destinations in the U.S., and (2) methods to empirically capture the impact of cultural positioning on individual health outcomes. Instead of relying on proxy measures of age at arrival and time in the U.S. to indicate where an individual is located on the acculturation spectrum, this study uses cultural consensus analysis to derive the substance and structure of a cultural model for *la buena vida* (the good life) among Mexican immigrant women in Birmingham, Alabama, and then assesses the extent to which respondents are aligned with the model in their everyday lives. This measure of ‘cultural consonance’ is explored as a moderating variable between age at arrival in the U.S. and number of depressive symptoms. Results demonstrate that for those who arrived at an older age, those with lower consonance are at the highest risk for depression, while those who are more aligned with *la buena vida* are at lower risk.

## 1. Introduction

Nearly 62 million Hispanic or Latino immigrants live in the United States, accounting for eighteen percent of the U.S. population [1]. As Mexican immigrants make up the single largest cohort of Hispanic/Latino immigrants, the health status of this group has important consequences for U.S. population health and for the health care delivery system. Research shows that among Hispanic/Latino immigrants in the U.S., Mexican women are at particularly high risk for developing depressive symptomology, suggesting that nativity and gender shape depression outcomes in critical ways [2,3,4,5,6]. It has long been presumed that culture also plays a role in shaping the lived experience of immigrants coming from different places and at different stages in their lives, though identifying the underlying mechanisms by which culture impacts health outcomes has been difficult to document empirically in social scientific research [7,8]. Acculturation denotes the process of transitioning from the cultural orientation of one’s upbringing to that of a host culture, and while this research paradigm is widely used to study immigrant health, it suffers from the lack of theoretical clarity around the relationship between culture and the individual as well as the lack of standardized methods to measure individual aptitude to a new cultural setting [9,10,11]. Instead, researchers often rely on proxy measures such as age at arrival and length of time living in the host country as indicators of cultural orientation or positioning. However, these do not capture the complexities of the acculturation experience as it is lived, nor do they account for the impact of socio-structural conditions on the health of individuals engaged in this process. As a result, studies on the relationship between acculturation and mental health trends have yielded inconsistent results, with the direction and magnitude of the effects varying considerably from study to study [12,13,14]. Innovative research is needed to gain a better understanding of how and why these factors influence risk of psychopathology for these newcomers and their children [8,15].

Rooted in biocultural anthropology, cultural consonance theory has emerged as a more empirically precise way of exploring the relationship between cultural realities and individual health outcomes. Defined as the “degree to which individuals, in their own beliefs and behaviors, approximate the prototypes for belief and behavior encoded in cultural models”, cultural consonance is associated with lower blood pressure, fewer depressive symptoms, and better immune-response functioning [16] (p. 3). In contrast to making assumptions about what members of a social group identify as desirable ways of living and being, this area of research uses a cultural models approach, which is designed to elicit these ideas from individuals in a particular social setting and to assess the extent to which they are shared within the community of reference. It is then possible to measure the degree to which individuals live up to these shared expectations in their everyday lives. When an individual is unable to carry out these culturally valued ways of being, most often due to social and structural constraints imposed on them, this may result in the feeling that life has not worked out the way that it is supposed to, an experience that may activate a stress response [17]. It is hypothesized that low cultural consonance adversely impacts mental health because individuals do not see themselves—and are not affirmed by others—as valued members of society. Cultural consonance provides a means of more precisely accounting for the relationship between culture and the individual and exploring the impact on health outcomes, while controlling for other variables known to influence those outcomes [18]. This approach allows for the integration of culture as an analytic variable in immigrant health research while not neglecting the impact of social, political and economic constraints that limit choice and movement for immigrants in the U.S.

The study described here was designed to better understand the attributes and organization of a cultural model for *la buena vida* (“the good life”) among Mexican immigrant women in Birmingham, Alabama, and how the ability to live up to this shared cultural model impacts risk of depression. Our conclusions suggest that cultural consonance with *la buena vida* moderates the relationship between age at arrival in the U.S. and depression. Individuals who immigrated at an older age and had low cultural consonance were the most likely to show symptoms of depression. Using an inductive, mixed methods approach, the research process rests on an empirical foundation that integrates and builds on the meaningful ways in which participants talk about and experience their world, including the structural and interpersonal barriers that condition their daily realities and impede their pursuit of *la buena vida.*


## 2. Background

A year after publishing a seminal article on acculturation [19], one of the authors noted that for the construct to be useful, the relationship between culture and its human carriers needed to be clarified [20]. To this day, research that considers the impact of acculturation on health has struggled to empirically demonstrate the relationship between culture—a property of the aggregate—and the individual, who variably internalizes and acts on the collectively held notions about how one ought to live and be in a particular social context [16]. In a 1960 article in social epidemiology, Cassel and colleagues advocated an open-system model in the study of disease, suggesting that the lived experience of cultural change comes to bear on health status in significant ways. Defining culture as the “fabric of meaning in terms of which people interpret their experience and guide their action”, their framework included social adjustment, which they defined as the “adequacy with which a person fulfills his social roles in terms of community expectation” [21] (pp. 945, 942). They hypothesized that for individuals living in an Appalachian town undergoing modernization, successful adjustment to changing expectations would be linked to better health outcomes, while the failure to adapt to these changes would be indicated by disease. Cassel’s approach is premised on the idea that individuals generally seek to conform to the influences and expectations of their social group. When they are able to demonstrate success vis-a-vis the collectively held notions of how to live and be, they are affirmed and validated in their social interactions, which bodes well for their health status. When individuals do not meet these expectations, this may provoke a sense of inadequacy or the feeling that life has not worked out as it is supposed to [22]. This experience may activate the stress response, the repeated arousal of which leads to greater risk of disorder and disease [23].

Antonovsky built on these ideas in developing salutogenic theory, which reaffirms the link between culturally associated stress and mental health outcomes [24]. In this framework, health promotion is facilitated by a sense of coherence, the feeling that life makes sense, that it is “comprehensible, manageable and meaningful” (p. 15). This sense of coherence depends on “one’s position in the social structure and on one’s culture” (p. 15). While Antonovsky notes the importance of culture, the measurement tool used to empirically test individuals’ sense of coherence is not culturally specific; rather, it is a generalized scale that essentially measures emotional responses to life situations. This assumes that everyone is working towards the same goals in life and that they will have similar responses when those are not met, obscuring the fact that both common goals and physiological responses are produced within a unique cultural space of meaning and manifest in different ways in different settings. 

More recently, Berry’s model of acculturation separates the group-level process of culture change from the individual-level psychological process of adapting one’s beliefs and behaviors to fit within a new cultural milieu [25]. Berry suggests that acculturative stress results when individual experiences do not conform to those of the cultural group or when one’s aptitude to a new cultural landscape is lacking. Subsequent research has found acculturative stress to be more proximate to psychological distress than acculturation, especially for those with low social support [26,27]. One study found that among international students at a U.S. university, those with broad social support networks and an integrative mode of acculturation (in which an individual embraces both their culture of origin and the new culture simultaneously) experienced lower levels of acculturative stress [27]. These results were supported among Korean immigrants in the U.S. and linked to fewer symptoms of depression [28]. In their exploration of potential links between social determinants and wellbeing [29], researchers found that mental wellbeing of Mexican mothers in the U.S. was positively associated with family cohesion and negatively associated with social isolation. They suggest that family cohesion can be disrupted by family members living outside the U.S., occupying different levels of acculturation (particularly regarding preferred language usage) and having different documentation statuses within the household, and that these conflicts lead to feelings of social isolation, which are associated with greater mental distress. In recognition of the multi-leveled factors that influence mental well-being, Riedel and colleagues developed an integrative framework that combines Berry’s acculturative stress model and Antonovsky’s salutogenic theory in an effort to better capture the impact of resistance resources and coping mechanisms during cultural adjustment for migrants [30]. These are associated with increased cultural competence and better person–environment fit. 

All of these frameworks are rooted in the psychosocial stress hypothesis, which suggests that individuals who are deprived of meaningful participation and validation in the wider society are at greater risk of poor health [31,32]. The crux of the problem with all of these theoretical frameworks is the lack of specification with regard to what actually matters to individuals, what they want their lives to look and be like, and how mental health suffers when their actual lives do not align with these ideals. According to Escobar and Vega, “the challenge…is to identify the necessary and sufficient dimensions of culture salient to cultural orientation. This specification problem is at the heart of confusion about the value and consistency of acculturation measurement” [9] (p. 737). Anthropologists have called for more precise models of culture and culture change that better account for the proximate person–environment dynamics that play out in daily lived experience for immigrants and elucidate the underlying pathways by which this experience impacts psychosocial stress outcomes [33]. This includes a serious consideration of how sociopolitical conditions shape the acculturation experience and are implicated in health status [15,34].

## 3. Theoretical Orientation: Cultural Consonance

In the large body of research on acculturation, one question that goes unanswered (and often unasked) is “acculturation to what?” Emerging from cognitive anthropology, the theory and methodology of cultural consonance resolves both the theoretical ambiguity and lack of empirical clarity regarding what culture is, how it drives ideas and behaviors about how to live and be, and why it matters in terms of health outcomes [16]. Grounded in a theory of cultural models [35] and methods that enable the quantification of those models through cultural consensus analysis [36], this approach provides an empirically satisfying way to infer the substance and structure of cultural knowledge relevant to a particular domain of life as well as the degree to which it is shared and valued within the community. Cultural models are stripped-down, skeletal representations of particular domains of life that include the elements that make up the domain as well as how those elements are linked together [16]. This approach is reminiscent of work by Rosch on “prototypes” [37], which applies to the meaning of words and the relationship among words in a particular semantic category, as well as Kronenfeld’s more general notion of “prototypicality” [38], which is extended to include actions and events. A cultural model denotes the best exemplar, or most prototypical, schema of a particular cultural domain. This can be inferred by measuring the degree to which knowledge about that domain is shared and agreed upon among individuals who are presumed to share a culture.

It is then possible to measure the extent to which individuals in that community are aligned with these collectively held models in their actual lives, be they models of family life [39], religious commitment [40], food [41], or broader models of life goals [22]. In turn, consonance with a salient cultural model can be explored as an influential variable on health outcomes while controlling for other variables that are known to influence those outcomes. Lack of consonance may be a function of the burdensome social and economic constraints that infringe on individuals’ abilities to put into motion the understandings that they share with their community about how to live a good life [17]. Cultural consonance research has demonstrated unequivocally that individuals who are unable to reproduce the shared expectations of what a person should be or should achieve are at greater risk for poor health outcomes, including depression [22]. This approach is similar to the work of García Martín on *el bienestar* [42], in which a “bottom-up” approach is used to identify the affective-cognitive dimensions that influence subjective well-being—health, sociodemographic variables, personal characteristics, behavioral variables and life events—as well as an individual’s assessment about their own well-being. Our research takes this a step further by considering how this alignment (or misalignment) between what is considered desirable and what life is actually like for individuals impacts measurable mental health outcomes. 

For the purposes of empirical validation, the methods of cultural consonance provide an analytically precise way of measuring culture in its aggregate form and exploring the direct relationship between culture and the individual. Once a meaningful and salient cultural domain is identified and the features relevant to that domain are elicited from participants, cultural consensus analysis (CCA) is used to systematically test for the presence of a cultural model for that domain. A cultural model is posited if there is a high degree of similarity in individuals’ responses to a set of standardized questions; this model represents the most likely set of “culturally correct” responses to the questions asked [43,44]. Individuals learn and internalize this information through social interaction and observation; what makes the models cultural is the fact that they are at least partially shared by members of a community and that they are directive with regard to motivating individual behavior. By analyzing the extent to which respondents agree on the importance of the features of the domain, cultural consensus estimates the best prototype of how a reasonably competent individual would rank the importance of the items. Referred to as the “answer key”, this estimation is used to construct a survey that asks respondents about their possession of the items in the model or adherence to a set of values or behaviors as delineated by the model. When individuals are not able to fulfill the expectations set forth by the cultural model, they see themselves and are seen by others as not able to achieve these widely shared life goals [17]. The resulting feeling of inadequacy is reinforced in social interactions where individuals are marginalized or disrespected, an experience that is known to activate the hypothalamus-pituitary-adrenal axis and sympathetic nervous system. These chronic assaults on the body lead to a higher allostatic load, which over a lifetime, lead to poor health [23,45]. For example, in a study of Hispanic immigrants in a southern town, Read-Wahidi and DeCaro found that immigrants who were more consonant with a cultural model of devotion to the Virgin of Guadalupe (a highly salient form of devotion in Mexico) were buffered from the effects of immigration stressors on psychological distress [46]. The current study looks at a similar population and considers how consonance with a more general domain of lifestyle impacts depressive symptomatology.

## 4. Materials and Methods

### 4.1. Participants and Sampling Procedures

Eligible participants were women over the age of nineteen who were born in Mexico and currently living in Birmingham, Alabama. While we acknowledge the rich diversity of Mexican culture and heritage, country of origin is a common analytic category in immigrant health research and appears to be a significant factor in health disparities within the Hispanic/Latino immigrant population in the U.S. [47]. Further, the focus is not on Mexican culture, per se, but on a cultural model that takes shape as women from all over Mexico converge in a relatively new immigrant destination in the U.S., where both daily activities and long-term goals are renegotiated and adapted to the new setting and where the ability to act on these goals is subject to a common set of social and structural constraints, regardless of documentation status. We limited the focus to women because Mexican women are twice as likely to develop symptoms of depression as Mexican men [2,3,4,5,6] and because gender shapes the migration and acculturation experience in important ways [2,48]. 

Four convenience samples of participants were recruited primarily through word of mouth, as well as through flyers at Hispanic grocery stores, churches, and service agencies. Semi-structured and structured interviews were conducted in Spanish and took place in the homes of participants and at two organizations that provide legal support and other services to the Hispanic/Latino community in Birmingham. Respondents provided consent and were asked to provide demographic and background information, including age, marital status, occupational status, household income, and educational level, as well as age at arrival in the U.S., number of years living in the U.S., and their self-reported level of proficiency in English. The research took place from May 2016 to June 2017. The Institutional Review Board at the University of Alabama approved this study and the sampling procedures used to recruit participants (IRB protocol #15-OR-292-R2).

### 4.2. Research Design

This study used an iterative sampling design where each phase of data collection was used to develop the next phase. The qualitative portion of the study consisted of a semi-structured interview with participants in the first sample that covered topics such as motivations for relocating to the United States, what the journey was like, how life in Alabama is different than in Mexico, and how participants are faring mentally and emotionally in their new location. These interviews were analyzed for themes and common sentiments, which were used to contextualize life as Mexican immigrant woman in Alabama and to ground the next phase of the research, which focused more explicitly on what *la buena vida* (“the good life”) means and how to achieve it. 

The quantitative research proceeded in two phases. In the first phase, cultural domain analysis was used to identify a salient domain of life within this community, to explore how meaning is constructed within that space, and to empirically capture how the domain is cognitively organized in individual minds [49]. In early discussions and interviews, it was clear that respondents use the term *la buena vida* to talk about a desirable life and the components that are necessary to achieve it. In the first sample (*n* = 31), participants engaged in a free list, in which they were asked to list the kinds of things that they believe are important or necessary for *la buena vida.* The free lists generated a long list of terms that was subsequently reduced to a set of the most frequently mentioned items. The final list included material possessions indicative of a modern, middle-class lifestyle, long-term goals pertaining to education and employment, and characteristics necessary for being a good person. 

These terms were written on blank notecards, and a second sample of respondents (*n* = 31) was asked to sort the terms into piles based on similarities or like qualities. During this pile-sorting activity, respondents were asked to talk through their reasoning for grouping certain items together. For example, a respondent might group refrigerator and oven and washing machine together under a category of “household appliances”. Alternatively, she might group refrigerator and car together as “things that are used daily”. These different ways of categorizing the items were noted. Using multi-dimensional scaling analysis [50], these data were analyzed to better understand the principal dimensions of meaning along which respondents tend to organize the domain. This results in a two dimensional “cognitive map” of the semantic space in which items that appear in the same pile more often will appear closer together on the map. This provides a sense of how most participants think about how these different items fit together. 

Next, we explored the way the different items are prioritized in people’s lives. A third sample (*n* = 41) was asked to rank order the items in terms of their importance in achieving *la buena vida*. To make this task easier, respondents first grouped the terms into four piles, ranging from very important to least important. Then, they ranked the items within each pile in order of importance. The result was a full rank ordering of the terms from the free list. Participants were reminded to rank the items based on their perception of how important they were to the community as a whole, not necessarily how they themselves would prioritize the items. Again, participants talked through their process of organizing the items and explained why they felt some items were more important than others. Using cultural consensus analysis [36], the rankings were analyzed to get a sense of how the attributes of the domain are prioritized in participants’ lives and the extent to which participants share ideas about what kinds of things are most important in achieving *la buena vida*. The first factor loadings generated by consensus analysis estimate the cultural competence of each individual, or the extent to which her rankings line up with the average rankings of the whole sample. This measures the degree of sharing between participants and provides a sense that the participants are not organizing the cards randomly but according to a shared schema. These collective responses are used to calculate an “answer key”, which provides the best estimate of how a reasonably culturally competent individual would likely rank the items in the domain [51]. The second factor loadings measure residual agreement, or patterned deviations in agreement between subsets of respondents [52].

In the final phase of the research, the ranked terms were incorporated into a cultural consonance survey that assessed the extent to which individuals align with the cultural model inferred from the cultural consensus analysis. This included questions regarding possession of material goods as well as alignment with longer term goals and desirable character traits. Items were differentially weighted depending on their relative importance according to the answer key. Possession of basic household items such as refrigerator or computer was asked as a series of yes/no questions, while items that depended on the respondent’s personal interpretation of her life circumstances or her personal character were asked about in a series of Likert-response statements with which respondents could agree not at all, a little, more or less, or definitely. For example, a question related to financial resources read: “I have enough money to buy the things I need”. These responses were recoded as dichotomous variables in the final analysis. Upon competition of the survey, each respondent was given a cultural consonance score, an indication of how closely she aligns with the collectively held model of *la buena vida.*
Figure 1 shows the step-by-step process of data collection. 

### 4.3. Outcome Measures and Analytic Strategy

To measure depressive symptomatology, we used the CES-D Spanish-language depression scale, a 35-item questionnaire designed to assess the major symptoms of depression, including depressed mood, changes in appetite or sleep, low energy, feelings of hopelessness, low self-esteem, and loneliness. Multiple regression analysis (SPSS v.23) was used to explore the extent to which cultural consonance moderates the strength and direction of the relationship between age at arrival in the U.S. and depressive symptoms, while controlling for the covariates of age, socioeconomic status, and time living in the United States. We hypothesized that cultural consonance with *la buena vida* would exert a more direct impact on depressive symptoms than typical measures of acculturation and that it would moderate the relationship between age at arrival and depression. 

## 5. Results

### 5.1. Qualitative Themes

Semi-structured interviews were designed to better understand what life is like for Mexican immigrant women in Alabama, their reasons for being here, their long-term goals, and the reality of their daily experiences. When asked about why they relocated to the United States, participants most commonly answered that they sought to improve the quality of their lives, to find work and save money (often to send home to family in Mexico), and so their children would receive a good education. Some women came to be with their partners who were already living and working in Alabama. Some cited the desperation to escape the violence in their hometowns, including abusive partners. For nearly all of the participants, relocating to the United States was a measure of last resort; they had no illusions about “the American Dream;” they only hoped for a better life than the one they had in Mexico, if not for themselves, for their children.

Most of the women I interviewed crossed the Mexico/U.S. border on foot, often guided by a *coyote* (human smuggler) that they had paid to lead them. For these participants, the journey was not easy. For some, it involved getting caught and sent back to Mexico or having to wait at checkpoints without knowing when or if someone would come for them. One participant described the journey as “*la más fea de mi vida”* (the worst or ugliest experience of her life). They described the difficulty of walking in the desert at night and slipping on rocks while crossing the river. Some were assaulted. Some hid in trunks of cars, terrified of being discovered by authorities. 

In addition to physical difficulties and injuries, the journey also took an immense emotional toll. All of the women I interviewed talked about how hard it was to leave their homeland, family members, churches, and in some cases young children. Even in their desperation for a better life, leaving *la patria* was not easy. They lamented the feeling of uncertainty that even if they made it across, they did not know what would become of their lives or if they would succeed in doing what they set out to do. 

Life in Alabama has not been easy for most of these women, if even some of the hardships that characterized their lives in Mexico have been relieved. While many enjoy a higher standard of living (running water, consistent electricity, functioning toilets), many long for the freedom they had in Mexico to move around without fear of being treated with hostility or being apprehended by authorities. Even in neighborhoods made up of mostly immigrants from Mexico, there was a sense of apprehension and distrust—a stark difference from the comfort of being in community and having people to rely on for support. One woman explained that in her small ranching community in San Luis Potosí, while they did not have much in terms of material possessions, “*la puerta se queda abierta*” (our door was always open), meaning she had an extended support system, with neighbors and relatives and friends constantly in and out of each other’s home and active in each other’s lives. “*Pero aquí les dicen que no abran la puerta a nadie, es peligroso*” (but here they tell you not to open the door to anyone, it is too dangerous), she said of life in Birmingham. 

A common sentiment expressed among older women was the change in family dynamics, specifically the weakening of a sense of *familismo* (familism). They expressed discomfort in not knowing their children’s friends or their parents, most often due to a language barrier. This posed a problem in communicating with their own children as well, as many had greater fluency in English and only spoke Spanish begrudgingly. This caused a profound sense of disconnect and discord in the homes—mothers frustrated with children for speaking to them in a language they did not understand, and children frustrated with mothers for not learning English. “They were young when they started with English. Me, I’m too old, and for me it just doesn’t stick”, one participant explained. The children’s proficiency in English did have its benefits, for example, if a mother needed a child to translate for her with a medical professional or other service agent. In another sense, however, this reversal of parenting roles was spoken of as a source of shame for the mothers and confusion for the children. 

Finally, participants expressed concern over deportation, for themselves or for family members. I interviewed one woman whose husband had recently been deported for the third time after he showed up to pay a drunk driving fine and was apprehended at the courthouse. Others talked about hearing reports on the radio that Immigration and Customs Enforcement (ICE) agents were setting up traffic blockades in an effort to round up as many undocumented immigrants as they could and send them back to Mexico. One woman expressed fear over going inside Walmart because she thought the policeman at the entrance might be an ICE official. This palpable sense of fear and anxiety became worse after the 2016 election because many assumed deportation efforts would be prioritized and intensified under the Trump administration. 

One of the overarching themes that emerged from this phase of research was the tension between longing for their old way of life and embracing a new life in a new place. For those with children, this tension manifested as a desire for the children to have a better education and more life opportunities in the U.S. but also feeling disconnected from them as they adopted a more Americanized way of living. Others discussed the tension between longing for a supportive community and struggling to find a sense of place and belonging in Alabama—both in terms of hostility from their reluctant hosts as well as feeling distrusting of other immigrants. A few eventually wanted to return to Mexico; others were resolved to stay the course in the United States. 

### 5.2. Sample Demographics 

Table 1 shows the descriptive statistics for all four samples recruited for the quantitative component of the study. The average age of all participants was 37. Age at arrival in the U.S. ranged from age 10 to age 60, with an average of 24, while number of years living in the U.S. ranged from 2 years to 40 years, with an average of 12 years. Fifty percent of participants arrived between 2004 and 2009. Nearly one-third of participants described their occupational status as *ama de casa* (housewife). Of those in the workforce, the most common jobs were in housekeeping and restaurants. For over half of the participants annual household incomes were between USD 10,000 and USD 15,000. Most participants had not advanced beyond primary school; only a few had some college experience. 

### 5.3. Cultural Domain and Cultural Consensus Analysis

Through interviews and participant observation, it became clear that women in this community often talk about *la buena vida* as something that they are pursuing, something that is meaningful to them and that they want for themselves and their families. The purpose of the first phase of quantitative data collection was to investigate *la buena vida* as a cultural domain, to empirically capture the substance and the structure of the domain in terms of what participants mean when they talk about *la buena vida* and the extent to which they share ideas about these meanings with others. Importantly, while many acculturation studies are deductive in that the researchers make assumptions about cultural positioning based on proxy measures, these methods are inductive, meaning that they allow for the elicitation of knowledge from respondents themselves. In the first sample, (*n* = 31), participants were asked a simple question: what kinds of things are necessary or important to achieve *la buena vida*? This generated a long list of items that included both material goods that characterize a modern, middle-class lifestyle (e.g., house, car, refrigerator, cell phone) as well as non-material items such as positive character traits (e.g., being positive, helpful to others) and long-term migration goals (e.g., good education for children, access to healthcare). After combining synonyms and alternative phrasings, the full list was reduced to a set of the 42 most commonly mentioned terms (see Table 2). 

To explore how respondents organize this cultural domain, a second sample of respondents (*n* = 31) was asked to sort the selected terms (written on notecards) into piles based on similarities or like qualities. Respondents are free to sort the terms however they see fit, as long as they do not group all the cards in one pile nor group all the cards in separate piles. These data were analyzed using multidimensional scaling (MDS) in order to better understand the principal dimensions of meaning along which respondents organize the domain. MDS arrays the terms at relative distance from one another depending on how often they are grouped together, which produces a graphical representation of how the semantic space is organized in participants’ minds. From this, the attributes that respondents are using to categorize the different items can be inferred [43]. The results indicated that participants tended to organize the domain along two primary dimensions—immediate needs verses long-term goals and material necessities or desirables verses non-material aspirations, both in a personal sense and for the family. Hierarchical cluster analysis in ANTHROPAC [49] verified that the short-term material items clustered into two subgroups, one consisting of a place to live, a car, and basic household items such as electricity, refrigerator, washing machine and hot water, and the other consisting of modern technology items and communication devices, such as cell phone, internet access, TV and computer. The non-material items can be broken down into long-term goals and leisure time activities. The items associated with long-term improvement of one’s station in life included getting a good education for the children, learning English and gaining access to healthcare, as well as items related to being a good person, such as being patient and positive, going to church and helping others. The final cluster consisted of leisure time activities such as rest, exercise, listening to music, being outside and spending time with friends. For further verification that respondents were not sorting the cards randomly but were indeed drawing from a shared understanding of how the terms should be logically grouped together, Borgatti suggests using cultural consensus analysis (CCA) to analyze the pattern of agreements among respondents in order to ensure that the pattern is “consistent with a single culture, rather than two or more conflicting groups” [49] (p. 275). While not a traditional factor analysis, CCA uses factor analytic methods to estimate the degree of sharing of cultural knowledge among respondents. If the ratio of the first to second eigenvalue is greater than three, this indicates that there is single pattern of responses [51]. Consensus analysis on the unconstrained pile sort data yielded an eigenvalue ratio of 10.4, suggesting that participants were largely in agreement with one another about how to conceptually categorize the various terms associated with *la buena vida*.

A third sample (*n* = 41) was asked to rank order the items in terms of their importance, and this was analyzed to get a sense of how the attributes of the domain are prioritized in women’s lives. Consensus analysis on the full rank ordering of the items yielded an eigenvalue ratio of less than three to one, indicating that there was no overall agreement regarding the importance of the elements (and rendering the competence values from the first factor loadings irrelevant). However, analysis of the second factor loadings—a measure of residual agreement—indicated that two subgroups of individuals did seem to agree more with one another than with the overall group. Dividing the sample into two subsets based on their proximity in the second factor loadings, both subgroups yielded an eigenvalue ratio of three or greater. As an aggregate, both groups ranked having a good job and having food to eat very high, though they differed in their prioritization of material items, particularly modern technology devices. One group (*n* = 27) deemed the material goods less important and prioritized items related to long-term goals and self-improvement, including better opportunities for children and positive character traits. This group tended to rank spending time with family, good education for the children, learning English, going to church and being positive as highly important. Items such as television, computer, internet access, and cell phone were ranked towards the bottom. One respondent emphasized the importance of items related to being a good person and fulfilling your duties as a mother and a person of faith, suggesting that these things “come first”. She characterized material items as what one needs to be comfortable in daily life, but she reinforced that simply having the material items is not enough to truly achieve *la buena vida.* On the other hand, the second group (*n* = 14) tended to be more concerned about the immediate needs of daily life, such as household goods, transportation and technological devices, as opposed to the more future-oriented aspects of improving one’s (and one’s family’s) position in life and being a good person. Table 3 shows how the two groups broke down by covariates. The subgroup with higher socioeconomic status and more proficiency in English clustered together in prioritizing basic household items and technological devices, and the subgroup with lower socioeconomic status and less English proficiency clustered together in their prioritization of items related to self-improvement and fulfilling familial and social obligations.

### 5.4. Cultural Consonance, Age at Arrival, and Depression

Cultural consonance with *la buena vida* was assessed using two different scales based on the residual agreement analysis, one that weighted material items and particularly technology devices more heavily and one that gave more weight to the long-term goals and abilities necessary to fulfill communal and familial duties expected of women in this community. Despite the apparent divergence in the articulation of priorities, cultural consonance tended to extend across both scales, indicated by the fact that the two sets of consonance scores had very strong significant correlations (r = 0.997). This suggests that both the materialistic items and the more idealistic items related to self-improvement and long-term objectives are encompassed in one culturally salient model of *la buena vida*. Further, while respondents may be diverging in the ways they articulate which items in the domain should be emphasized or prioritized in order to achieve *la buena vida*, those who are consonant tend to be consonant with both versions of the model. Another way of looking at this is that women who have managed to acquire the indicators of a middle-class lifestyle are more likely to see themselves as good in a moral sense, while women who do not have access to these material desirables are more likely to see themselves as unable to fulfill their familial and social duties as women. This is an interesting finding, a discussion of which is beyond the scope of this paper [53], but it is mentioned as a justification for using a composite scale of cultural consonance that aggregated the two differentially weighted scales.

An analysis of bivariate correlations (see Table 4) indicates that younger age at arrival (r = 0.247, *p* = 0.039) and better command of English (r = −0.295, *p* = 0.013) were associated with fewer depressive symptoms. Age, socio-economic status, and time spent living in the U.S. did not have significant bivariate correlations with depressive symptoms. Cultural consonance was shown to have an inverse relationship with number of depressive symptoms, confirming the hypothesis that greater consonance is linked to better mental health outcomes. The next phase of the analysis was to look at the interaction between cultural consonance and typical measures of acculturation. 

To explore these relationships further, we conducted a hierarchical multiple regression analysis of depressive symptoms in which standard covariates are entered first (age, socioeconomic status), followed by acculturation measures (age at arrival, length of time living in the U.S. and English proficiency) followed by cultural consonance with *la buena vida* (see Table 5). The interaction effect between age at arrival and cultural consonance is entered next. Standardization was used to deal with collinearity in the interaction terms and in order to show the change in depressive symptoms with one standard deviation increase or decrease in the independent variable. The dependent variable, depressive symptoms, remains in the original metric. The interaction between age at arrival and cultural consonance is a significant predictor of depressive symptoms (B = −0.31, *p* = 0.01), suggesting that the strength of the impact of cultural consonance on mental health changes depending on age at arrival. 

Figure 2 shows that among those who immigrated to the United States at a younger age (1 s.d. below the mean), depressive symptoms are low and there is virtually no significant interaction with cultural consonance. However, for those whose age at arrival is closer to the mean, not only are depressive symptoms higher, but they are significantly higher for respondents with lower levels of cultural consonance. This effect is even stronger for those whose age at arrival is one standard deviation above the mean, indicating that the combination of being older upon arrival and having low consonance puts one at the greatest risk for experiencing more depressive symptoms. On the other hand, for those who were older upon arrival and have managed to achieve higher levels of cultural consonance, their risk of experiencing depressive symptoms is basically equivalent to respondents who were younger upon arrival. 

## 6. Strengths and Limitations

Those accustomed to working with large data sets may find the small sample sizes used here concerning. It is important to emphasize that a cultural models approach is designed to capture information—the purpose is to sample knowledge, not people. Weller and colleagues have recently shown that when collecting knowledge about a particular domain through open-ended free listing, “saturation in salience” can be reached with a relatively small sample size [44]. They suggest that as long as the domain itself is salient within a community of reference, very little new and important information about the domain is captured beyond thirty free lists. That said, cultural domain analysis only tells us about a particular point in time. More research is needed on if and how cultural models change over time, especially if models change in ways that make them more or less attainable for certain individuals. We also need more research on models of lifestyle in other cultural groups, migrants and otherwise, to ensure that the models are specific to the community in question. In exploring the moderating effect of cultural consonance, the small sample size is dealt with by only including the most important variables of interest in the regression models. There are, of course, many factors that may be important here, but our purpose was to show that beyond age and socioeconomic status, cultural consonance does influence mental health *and* that it moderates the negative impact that older age at arrival in the U.S. otherwise has on depressive symptoms. 

## 7. Discussion

The study described here considers the relationship between cultural consonance with a model of *la buena vida* and depressive symptomology among Mexican immigrant women in a southern U.S. town and explores the extent to which consonance moderates the relationship between age at arrival and mental health status. What sets this approach apart from other studies of immigrant health is that it centralizes the perspectives and opinions and understandings of the community under consideration, rather than assuming a priori what is important or necessary for individuals in that community. In addition to eliciting this information from community members, the qualitative interview data provide insight into what kinds of barriers these women face in carrying out what they understand *la buena vida* to be. These are contextualized by the geographical location of the new setting (a relatively new immigrant destination in the U.S. South), the attitudes of the host community, and the structural conditions that limit choice, movement, and access to services. Rooted in a biocultural perspective, this research seeks to elucidate how the lived experiences of individuals embedded within a particular cultural setting and positioned within a particular life-history stage contribute to noted epidemiological trends and health disparities, particularly the increased risk of depression among Mexican women in the U.S. 

There is little to discuss regarding the qualitative themes because they speak for themselves. The point of summarizing these sentiments about what life is life for Mexican immigrant women in Alabama is to contextualize and ground the systematic data collection and analysis that followed and to ensure that participants’ voices were centralized throughout the study. While epidemiological studies usually invest more resources in obtaining large samples, they typically do not ensure that the measures they are using are culturally valid within a specific community. A mixed-methods approach invests more in obtaining variables with high “emic validity” in order to better understand the process of acculturation as it is experienced [54]. This is an effort to “resocialize epidemiology” by bringing some clarity to known statistical patterns by investigating the relationship between cultural experiences and health outcomes in meaningful ways [55]. 

We use a cultural models approach to better understand how participants conceptualize *la buena vida*. This approach can be distinguished from other work on prototypicality because it asks individuals to conceptualize an ideal, what they want their lives to be like, which may or may not align with their actual lives. The items mentioned in the free lists of what kinds of things are important or necessary to have *la buena vida* included material items indicative of a modern, middle-class lifestyle as well as character traits related to being a good person, particularly a good mother. This serves as a contrast to what Kaja Finkler describes as *la mala vida* (the bad life), which characterizes people (usually women) who are both economically and morally destitute [56]. It may be surprising to some readers that things such as safety, security and long-term immigration goals (i.e., gaining citizenship) were not mentioned often in the free lists. It seems that participants were thinking about *la buena vida* more in terms of what they can reasonably achieve or actualize in their everyday lives and less so in terms of governmental processes or policy changes that are necessary to ensure their citizenship or safety and protection under U.S. law. As mentioned in the qualitative themes, issues of insecurity and fear of deportation are central to the participants’ lives, so all of the results of the quantitative component of this study should be analyzed and understood in this overarching context. 

Salience in the free lists was achieved, and there was strong consensus regarding how to categorize the items in the domain. However, there was some disagreement among participants in the third sample over how to rank order the items. It was interesting that respondents who were better off economically prioritized the material goods while the less well-off emphasized items related to *being* good. It is as if women with fewer resources have an idea that as long as they remain “good” people, everything else will work out eventually. In listening to participants explain their process of ranking the items, it is clear that achieving some level of financial security is important but that to truly achieve *la buena vida*, one must also be able to demonstrate that she is committed to her family and her faith as well. While one would not expect to see this deviation in agreement in describing a salient and well-known cultural model, this makes sense in the context of the ever-present tension participants describe as characterizing their lives as migrants in the U.S. This is also verified in that even when the consonance scales were weighted differently, participants who were consonant with one version of the cultural model were highly likely to be consonant with the other as well. This leads us to the conclusion that *la buena vida* is a multi-centric cultural model. 

To our knowledge, this is the first study to examine cultural consonance as a moderator of the relationship between age at arrival and depressive symptoms for Mexican immigrant women in the U.S. The results suggests that for Mexican women who immigrated to the U.S. at an older age, meaning they spent more time in their country of origin, lack of cultural consonance with a collectively held standard of *la buena vida* puts them at heightened risk for developing symptoms of depression compared to those who have managed to actualize the cultural model in their own lives. To borrow the terms from Brown and Harris, the moderation effect shows that cultural consonance acts as a “provoking agent” that exerts a direct effect on risk of depression risk, while age at arrival in the U.S. serves as a “vulnerability factor” that puts certain people at greater risk of psychopathology depending on their ability to successfully carry out shared ideals about how to live and be [57]. It is interesting that we found no significant association between socio-economic status and depressive symptoms. This indicates that being culturally consonant requires more than resources and material goods. The lack of a significant association between age or time living in the U.S. suggests that these measures are not meaningful in themselves; what matters is why happens during this time and how it is experienced.

## 8. Conclusions

Despite the fact that acculturation is generally regarded as a complex and multidimensional process that cannot be transposed into linear form, unidimensional models of acculturation remain the most widely used in large-scale immigrant health studies [8]. While numerous studies have demonstrated a statistically significant association between acculturation and health outcomes, little research has sought to describe how cultural value systems are meaningfully structured, why they might be protective or dysfunctional for certain immigrant groups, and why and how they change over time. As proxy variables of acculturation, age at arrival and length of time living in the U.S. tell us very little about cultural orientation and even less about lived experience, so the association with health outcomes is not well understood. A cultural consonance approach is better suited to describe the substance and structure of salient cultural models as they take shape in the context of migration and resettlement and to consider how the ability to demonstrate success vis-a-vis these shared models is relevant to health and well-being. Studies, including this one, show that this matters a great deal in terms of mental and emotional wellbeing. To this end, the methods used here provide a way to bridge the gap between the epidemiological research that focuses on the relationship between proxy measures of acculturation and health outcomes with the social determinants research that situates health and wellbeing in the broader political-economic and social conditions that shape daily realities for Mexican immigrants in the United States. Exploring the meanings and the significance of *la buena vida* and the extent to which these ideas are shared within among Mexican immigrant women in Alabama offers greater insight into how these women understand their world and what they are working towards in this life. We can then consider how the ability to achieve consonance with a culturally valued lifestyle impacts their mental health. This research contributes to social scientific work aimed at enhancing our theoretical understanding of what culture is, how it functions in our lives, and why it matters in terms of health and wellbeing.

## Figures and Tables

**Figure 1 ijerph-19-05342-f001:**
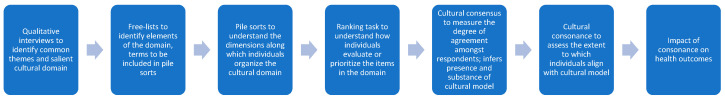
Step-by-step data collection and analysis process.

**Figure 2 ijerph-19-05342-f002:**
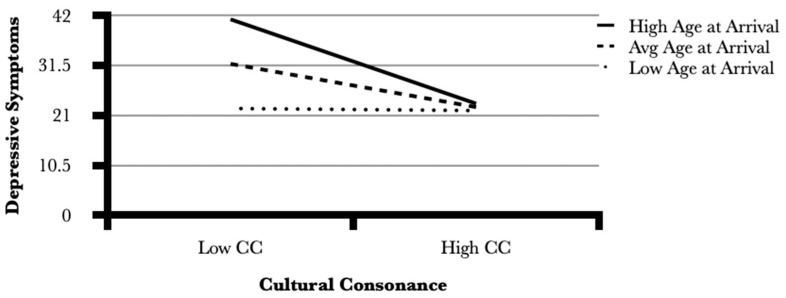
Interaction effect of age at arrival in U.S. and cultural consonance on depressive symptoms.

**Table 1 ijerph-19-05342-t001:** Descriptive statistics for Samples 1–4.

	Sample 1 (*n* = 31)	Sample 2 (*n* = 31)	Sample 3 (*n* = 41)	Sample 4 (*n* = 70)
Age	36.52 (19–70, 12.73)	40.13 (27–58, 9.09)	34.81 (19–54, 8.27)	37.44 (19–66, 10.07)
Age at Arrival	25.03 (10–60, 11.06)	27.03 (12–51, 8.69)	20.68 (6–48, 8.29)	23.43 (2–48, 9.69)
# of Years in U.S.	12.39 (2–40, 7.12)	12.87 (2–36, 8.09)	13.83 (2–29, 4.50)	13.31 (2–29, 5.02)
Occupational Status *	0.74 (0–3, 0.81)	0.29 (0–2, 0.59)	0.56 (0–3, 0.91)	0.61 (0–3, 0.91)
Household Salary **	1.26 (0–3, 0.86)	1.26 (0–3, 0.73)	1.12 (0–3, 0.64)	1.11 (0–3, 0.94)
Education ***	1.58 (0–3, 1.21)	1.55 (0–3, 1.03)	1.36 (0–3, 0.96)	1.29 (0–3, 1.00)
English Proficiency ****	1.29 (0–3, 1.24)	1.16 (0–3, 0.82)	1.29 (0–3, 0.99)	0.94 (0–3, 0.92)

* Occupational status measured as (0) does not work outside home, (1) unskilled worker, (2) skilled worker, (3) professional/business owner; ** Average weekly salary of household measured as (0) <USD 300, (1) USD 300–600, (2) USD 600–1000, (3) >USD 1000; *** Highest level of education measured as (0) no school or primary school, (1) secondary school, (2) preparatory school, (3) university; **** Self-assessed English proficiency reported as (0) none, (1), a little, (2) good, (3) very good.

**Table 2 ijerph-19-05342-t002:** Free-list terms for *la buena vida*.

Term	Frequency	Average Rank	Salience
Casa (House)	17	2.35	0.47
Tiempo con familia (Family time)	17	4.12	0.37
Coche (Car)	16	2.94	0.40
Positiva (Be positive)	13	6.77	0.14
Buen trabajo (Good job)	10	2.60	0.27
Dinero (Money)	9	4.33	0.19
Comida (Food)	8	5.13	0.15
Tiempo para estudiar (Study time)	6	6.17	0.09
Ropa (Clothes)	5	7.00	0.06
Aprender inglés (Learn English)	5	5.00	0.09
Educación para niños (Education for children)	5	7.6	0.05
Refrigerador (Refrigerator)	5	5.2	0.10
Internet (Internet)	4	3.00	0.09
Acceso a medicina (Affordable medicine)	4	9.75	0.02
Cuidado de salud (Health insurance)	4	6.75	0.05
Amigas (Friends)	4	6.50	0.05
Ayudar a otros (Help others)	4	6.50	0.04
Religiosa (Be religious)	4	4.50	0.08
Rezar (Pray)	3	9.67	0.02
Humilde (Modesty)	3	6.33	0.05
Horno/estufa (Oven/stove)	3	4.00	0.07
Tiempo libre (Free time)	3	5.67	0.06
Televisor (Television)	3	6.33	0.05
Ejercicio (Exercise)	3	6.67	0.03
Celular (Cell phone)	3	4.33	0.06
Ser amable (Be kind)	2	8.00	0.02
Ser espiritual (Be spiritual)	1	5.00	0.02

*n* = 1; Average response length = 8.74; Range = 5–13; Total items listed = 85.

**Table 3 ijerph-19-05342-t003:** Cultural consensus analysis by subgroup (Sample 3).

	Group 1 (Priority: Long-Term Goals)	Group 2 (Priority: Material Items)
# of Respondents	27	14
Age	36.74 (22–54, 8.34)	32.07 (19–43, 6.78)
Age at Arrival	22.52 (9–48, 8.62)	17.77 (6–28, 6.31)
# of Years in U.S.	14.26 (6–29, 4.39)	13.14 (2–21, 4.93)
English Proficiency *	1.93 (1–3, 0.78)	2.86 (1–4, 1.03)
SES **	4.00 (2–6, 1.21)	5.31 (4–7, 1.12)
# of Negative Competence Scores	1	2
Average Competency (range, s.d.)	0.50 (−0.36–0.87, 0.26)	0.54 (−0.21–0.90, 0.36)
Eigenvalue ratio	3.06	3.01

* Self-assessed English proficiency reported as (0) none, (1) a little, (2) good, (3) very good; ** SES measured as average weekly salary of household (0–3) and highest education level completed (0–3); See Table 1.

**Table 4 ijerph-19-05342-t004:** Bivariate correlation matrix between acculturation measures, cultural consonance, and depressive symptoms.

	Age at Arrival	Years in U.S.	English Proficiency	Depressive Symptoms	Cultural Consonance
Age at Arrival		−0.27 *	−0.24 *	0.25 *	−0.30 **
Years in U.S.			0.06	−0.09	−0.29
English Proficiency				−0.30 **	0.48 **
Depressive Symptoms					−0.30 *

* Significant at 0.05 level; ** Significant at 0.01 level.

**Table 5 ijerph-19-05342-t005:** Regression models of age at arrival and cultural consonance on depressive symptoms.

	Model 1	Model 2	Model 3	Model 4
Age	0.19	0.07	0.11	0.21
SES	−0.16	−0.07	0.14	0.18
Years in U.S.	- - -	−0.07	−0.11	−0.22
English Proficiency	- - -	−0.22	−0.19	−0.21
Age at Arrival	- - -	0.13	0.04	−0.70
Cultural Consonance	- - -	- - -	−0.29	−0.22
Age at Arrival × Cultural Consonance	- - -	- - -	- - -	−0.31 *
	R^2^ = 0.04; *p* = 0.10	R^2^ = 0.06; *p* = 0.10	R^2^ = 0.08; *p* = 0.07	R^2^ = 0.15; *p* = 0.01

*N* = 70; All variables (except dependent) standardized; * = significant at 0.01 level.

## Data Availability

This data set is not publicly available because it is still under analysis.

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
