# Peer review of "Age at Arrival and Depression among Mexican Immigrant Women in Alabama: The Moderating Role of Culture"

_ijerph, 2022, doi:10.3390/ijerph19095342_

Round 1
Reviewer 1 Report
A decent paper that claims to look at acculturation and health. I found the idea of using la buena vida interesting but I was a bit concerned about the bias that seemed to focus more on material items and leisure as opposed to what might be useful to think of as a subcategory within la buena vida: bienestar. I was surprised to not see security, safety from violent situations or other indices given the context many women are coming from. Crucially, there was also no discussion, given some of these women are clearly illegal in the US, of longterm citizenship goals or discussions around green cards and how that correlates to la buena vida y bienestar, surely some of these women must think about their legal status as connected to la buena vida in some way or another. I was surprised that this was not anywhere in the paper. In discussing acculturation and health it is essential to think about how welcome someone feels in the new culture, belonging and issues around being able to feel integrated are deeply related to mental health as well as physical: Sullivan, Stacciarini, Wong, Salas et al., are authors who should be read on this topic which needs to be considered before this paper can be published.
Author Response
Dear Reviewer,
Thank you so much for taking time to read our paper and offer helpful insights and suggestions. We’re pleased that you were satisfied with our literature review, the description of our methods, and the presentation of our results.
To address your concerns, we have made several revisions and additions, which are enumerated below.
1.) We agree that the concept of bienestar is relevant to our study and that it fits within the overarching category of la buena vida. In our revised draft, we mention the similar approach between cultural consonance and bienestar: “This approach is similar to the work of García Martín on el bienestar [51], in which a “bottom-up” approach is used to identify the affective-cognitive dimensions that influence subjective well-being – health, sociodemographic variables, personal characteristics, behavioral variables and life events – as well as an individual’s assessment about their own well-being. Our research takes this a step further by considering how this alignment (or misalignment) between what is considered desirable and what life is actually like for individuals impacts measurable mental health outcomes.”
As a point of clarification, we note that our study is built around a cultural model of la buena vida because that is the term most commonly used by our participants, and using an inductive approach, we wanted to ensure that the language of our study was reflected in the language used by participants.
2.) We also agree that security, safety from violence, and long-term citizenship goals are important in the everyday lived experience of our participants. These things did come up in semi-structured interviews and centered around the fear of deportation (see pg.#, lines #). Most of the long-term goals mentioned by participants centered more around their children than around themselves. In the free-listing activity (phase 1 of the quantitative component), when participants were asked what kinds of things are necessary or important to achieve la buena vida, the focus was more on material goods and leisure time activities as well as the attributes of being a good person. This seems to come from a place of what they can reasonably achieve for themselves as opposed to things that depend on governmental processes or policy changes. To address this concern, we added: “It may be surprising to some readers that things like safety, security and long-term immigration goals (i.e., gaining citizenship) were not mentioned often in the free lists. It seems that participants were thinking about la buena vida more in terms of what they can reasonably achieve or actualize in their everyday lives and less so in terms of governmental processes or policy changes that are necessary to ensure their citizenship or safety and protection under U.S. law. As mentioned in the qualitative themes, issues of insecurity and fear of deportation are central to the participants’ lives, so all of the results of the quantitative component of this study should be analyzed and understood in this overarching context.”
3.) Thank you for the suggestion to read up on the work of Sullivan, Stacciarini, Wong, and Sales. We mention the important work of these researchers in the background section: “More recently, Berry’s model of acculturation separates the group-level process of culture change from the individual-level psychological process of adapting one’s beliefs and behaviors to fit within a new cultural milieu [25]. Berry suggests that acculturative stress results when individual experiences do not conform to those of the cultural group or when one’s aptitude to a new cultural landscape is lacking. Subsequent research has found acculturative stress to be more proximate to psychological distress than acculturation, especially for those with low social support [52, 53]. One study found that among international students at a U.S. university, those with broad social support networks and an integrative mode of acculturation (in which an individual embraces both their culture of origin and the new culture simultaneously) experienced lower levels of acculturative stress [53]. These results were supported among Korean immigrants in the U.S. and linked to fewer symptoms of depression [54]. In their exploration of potential links between social determinants and wellbeing [55], researchers found that mental wellbeing of Mexican mothers in the U.S. was positively associated with family cohesion and negatively associated with social isolation. They suggest that family cohesion can be disrupted by family members living outside the U.S., occupying different levels of acculturation (particularly regarding preferred language usage) and having different documentation statuses within the household, and that these conflicts lead to feelings of social isolation, which are associated with greater mental distress. Riedel and colleagues developed an integrative framework that combines Berry’s acculturative stress model and Antonovsky’s salutogenic theory in an effort to better capture the impact of resistance resources and coping mechanisms during cultural adjustment for migrants [26]. These are associated with increased cultural competence and better person-environment fit.”
Reviewer 2 Report
This paper presents results of a mixed-method study of cultural fit and mental health among Mexican immigrant women in Alabama. Strengths of this paper include strong theorization and an innovative methodology. These are some fascinating results; I have only a few comments.
- You mention that there are bivariate correlations between symptoms of depression and some of the model variables, but these are not reported. A correlation matrix would be informative to include.
- It seems surprising that none of the other model variables reach the level of significance in any model (e.g., SES). Can you comment on this outcome?
- There seems to be a discrepancy between the table and the text in terms of the reported coefficient and significance level for the interaction (-0.31, p < .01 vs. -0.26, p = .05) unless there is something I’m misunderstanding here. Please check these.
- The construct of cultural coherence has some affinities with the social psychological construct of prototypicality – i.e., group members derive more benefits, including in terms of mental health, from their group identification to the extent that they are perceived (by themselves and by other group members) to embody the ideals of the group. While there are undoubtedly important distinctions between these concepts, it would be interesting to briefly tie these two literatures together in the introduction and/or discussion.
Author Response
Dear reviewer,
Thank you so much for taking the time to review our paper and for offering helpful suggestions on how to improve it. We are pleased to learn that you were satisfied with our literature review, our research design, the description of our methods, the presentation of our results and our conclusions. Thank you!
Again, thank you for your comments on how to improve our work. Below we enumerate how we have addressed each of your concerns.
1.) We added the bivariate correlations to our results: “An analysis of bivariate correlations indicates that younger age at arrival (r= 0.247, p= 0.039) and better command of English (r= -0.295, p= 0.013) were associated with fewer depressive symptoms. Age, socio-economic status, and time spent living in the U.S. did not have significant bivariate correlations with depressive symptoms.” We also added a correlation matrix (See Table 3).
2.) Yes, we found this surprising as well and imagine other readers might as well. We have added this explanation to the discussion: “It is interesting that we found no significant association between socio-economic status and depressive symptoms. This indicates that being culturally consonant requires more than resources and material goods. The lack of a significant association between age or time living in the U.S. suggests that these measures are not meaningful in themselves; what matters is why happens during this time and how it is experienced.”
3.) Thank you for pointing this out! The table is correct (the text was from an earlier model in which variables were not standardized). We have made the correction in the text (pg.12, Table 4).
4.) We agree that there is overlap with the construct of prototypicality. We included a reference to this in the background section in our discussion of cultural models: “Cultural models, are stripped-down, skeletal representations of particular domains of life that include the elements that make up the domain as well as how those elements are linked together [16]. This approach is reminiscent of work by Rosch on “prototypes,” [56] which applies to the meaning of words and the relationship among words in a particular semantic category, as well as Kronenfeld’s more general notion of “prototypicality,” [57] which is extended to include actions and events. A cultural model denotes the best exemplar, or most prototypical, schema of a particular cultural domain. This can be inferred by measuring the degree to which knowledge about that domain is shared and agreed upon among individuals who are presumed to share a culture.”
We bring it up in the discussion as well: “We use a cultural models approach to better understand how participants conceptualize la buena vida. This approach can be distinguished from other work on prototypicality because it asks individuals to conceptualize an ideal, what they want their lives to be like, which may or may not align with their actual lives.”
Thank you again for taking the time to review our work. Please let us know if there are additional clarifications to be made.
Reviewer 3 Report
The authors present an interesting and broadly articulated work. I have a few concerns that I have listed below.
The authors present an interesting and broadly articulated work. I have a few concerns that I have listed below.
The overview of the study needs to be clearer and more detailed. It is difficult for the reader to get a clear frame of all the steps. A chart would be helpful.
The hypotheses should be more precise.
A table with the items used is missing.
Author Response
Dear Reviewer,
Thank you so much for taking the time to review our work and offer helpful suggestions on how to make it better.
We are pleased that you were interested in our study and found the description of our methods, the presentation of our results and our conclusions satisfactory.
1.) Based on your comments and in order to improve our manuscript, we have attempted to make the overview of the study more clear by adding a chart of all the steps (Figure 1, pg.7).
2.) We have made our hypothesis more precise: “We hypothesized that cultural consonance with la buena vida would exert a more direct impact on depressive symptoms than typical measures of acculturation and that it would moderate the relationship between age at arrival and depression.” Thank you for this suggestion.
3.) We have added a table with the terms from the free list (Table 2, pg.10). Thank you for this suggestion.
Thank you again for your insights into how to improve our paper. Please let us know if there are additional clarifications to be made.
Round 2
Reviewer 3 Report
The manuscript seems to have improved. I would suggest that the authors remove several typos throughout the article.